# Influence of Vibration Dampers on the Vortex-Induced Force and Flow Characteristic of Deep-Water Jacket Pipe

Chao Luo [1], Zhirong Wei [2], Jiajia Chen [1], Liqin Liu [2,*] and Yongjun Yu [2]

1    Offshore Oil Engineering Co., Ltd., Tianjin 300461, China
2    State Key Laboratory of Hydraulic Engineering Simulation and Safety, Tianjin University, Tianjin 300072, China
*    Correspondence: liuliqin@tju.edu.cn

**Abstract:** Vibration dampers are widely used in power transmission line vibration reduction. In order to use them for wind-induced vortex-induced vibration (VIV) suppression of jacket pipes, the effect of the vibration dampers on the vortex-induced force is studied using the computational fluid dynamics (CFD) method. The range of Reynolds numbers in simulations is in the critical interval, and the Reynolds-averaged Navier–Stokes (RANS) equations and shear stress transport (SST) $k$-$\omega$ turbulence model are used to calculate the pipe with vibration dampers. The lift coefficient of the pipe is reduced by about 65% after the vibration dampers are installed. The effect of vibration dampers on the lift force and drag force is little affected by the change of wind speed. The same number of vibration dampers are installed in two rows, and the vortex shedding frequency is reduced by about 16% compared with that for one row. The vibration dampers destroy the wake vortex of the high-velocity areas around the pipe, thereby reducing the pipe's lift coefficient and the vortex-induced force. The vibration dampers have no obvious influence on the vortex far from the pipe.

**Keywords:** deep-water jacket pipe; wind-induced VIV; high Reynolds number; vibration damper; flow field influence analysis

## 1. Introduction

Deep-water jacket platforms are usually constructed horizontally on land and are mainly affected by wind loads in the horizontal direction. When the wind flows in the direction perpendicular to the pipe, it will not only generate drag force in the downstream direction, but also form vortices on the surface of the structure. The alternating appearance of the vortex makes the pipe vibrate transversely; that is, wind-induced vortex-induced vibration (VIV). During the construction of the jacket, some pipes are in a state of unilateral restraint, the natural frequency is low, and the wind speed at which vortex-induced resonance occurs is also reduced. Therefore, the influence of wind-induced VIV should be considered.

Many scholars have studied the problem of flow around a cylinder considering a high Reynolds number. Muk Chen Ong et al. [1] used the standard $k$-$\varepsilon$ model to simulate the two-dimensional flow around a cylinder with $Re = 10^6$ and compared it with the calculation results of the large eddy simulation. They found that the standard $k$-$\varepsilon$ model can better predict the flow around the cylinder in the critical and supercritical areas. Haixuan Ye et al. [2] used the SST $k$-$\omega$ model combined with the overset grid to simulate the two-dimensional flow around the cylinder in the range of $Re = 6.31 \times 10^4$~$7.57 \times 10^5$ and compared the experimental results, which proved the feasibility of using the overset grid to study the VIV problem. Kang et al. [3,4] modified the SST $k$-$\omega$ model of Jauvtis and Williamson [5], which confirmed that the modified model has better simulation performance. Ma et al. [6] measured the wind pressure distribution along the length of the cylinder and the lift coefficient of the cylinder in the range of $Re = 10^5$~$4.6 \times 10^5$ in the wind tunnel test. They

found that the flow state will change along the length of the cylinder, and the separation and reattachment of the bubbles can cause unstable changes in lift and pressure.

Vortex-induced vibration suppression methods can be divided into two categories: passive suppression methods and active suppression methods. The passive suppression method can suppress vortex shedding by changing the surface shape of the structure or installing accessories to change the flow field distribution of the structure, or structural vibration can be directly suppressed by installing vibration dampers and shock absorbers. The passive suppression method is simple to design and implement and has low cost, but it cannot effectively respond to changes in the external environment in time [7]. The active suppression method uses real-time monitoring of structural vibration response and flow field and actively interferes with structural vibration and flow field distribution so as to suppress vibration. The active suppression method can adjust control strategies and parameters according to environmental changes and has strong adaptability to the environment. However, active suppression requires external energy input, complex technology, and high cost. These suppression devices have been widely used in offshore pipelines, but there are few suppression devices used on jacket pipes.

Naoaki et al. [8] studied the flexible riser with helical strakes through experimental methods. From the change of cross-flow vibration frequency, it was concluded that the helical strakes can well suppress the occurrence of vortex-induced vibration. Han et al. [9] simulated the suppression effect of water drop fairing and short tail fairing on vortex-induced vibration by using the SST model and the URANS method modified by the arbitrary Lagrangian–Eulerian (ALE) method. The numerical results obtained were the same as the experimental results. Abdi et al. [10] simulated the cylinder covered with one, two, and three horizontally connected partition plates by fluid–structure interaction using the ALE method; they studied the vortex shedding frequency and the fluid force acting on the cylinder and conducted a comprehensive parameter study to determine the optimal layout of the plates with the maximum drag reduction and maximum vortex shedding frequency. Tulsi et al. [11] studied the flow-induced vibration of a cylinder with a rigid splitter plate and determined that the vibration amplitude of a cylinder with a rigid splitter plate is significantly smaller than that without a rigid splitter plate. When the plate length increases, the vibration amplitude decreases, and when the plate length is smaller, the vibration frequency is greater.

The vibration damper is a widely used vibration damping device in the power industry, which is easy to install and low in cost. In recent years, vibration dampers have been used in deep-water jackets to suppress vortex-induced vibration of structures. Qin used CFX software to carry out numerical simulation, then calculated and analyzed the force and flow field changes of the pipe after installing the "dog bone" with $Re = 200$. They concluded that the vibration dampers can effectively reduce the pipe lift coefficient. However, the Reynolds number considered in this study was low, while the diameter of the deep-water jacket pipe was large, which was in the high Reynolds number range during vibration.

In this paper, the wind-induced VIV of the deep-water jacket pipe under the condition of a high Reynolds number is studied, based on the CFD method. The influence of the two types of vibration damper arrangements on the lift coefficient and drag coefficient is also analyzed. The influence of the vibration dampers on the lift and drag coefficient after changing the wind speed in the interval of vortex-induced vibration is calculated. According to the vorticity contour, the reason for the restraining effect of the vibration dampers on the vortex-induced force is analyzed from the perspective of the flow field. This paper can provide a reference for the suppression of wind vortex-induced vibration of a deep-water jacket pipe.

## 2. Basic Theory of CFD Simulation

### 2.1. Governing Equation

The governing equation of incompressible viscous fluid flow is:

$$\nabla \cdot u = 0 \tag{1}$$

$$\frac{\partial u}{\partial t} + (u \cdot \nabla)u = -\frac{1}{\rho}\nabla p + v\Delta u \tag{2}$$

where $u = (u, v, w)^T$ is the velocity vector of the flow, $p$ is the pressure, $\rho$ is the fluid density, and $v$ is the kinematic viscosity coefficient of the fluid.

The time-averaged method examines the effects of pulsation by viewing turbulent motion as a superposition of time-averaged flow and instantaneous pulsating flow. The Reynolds averaging method [12] was introduced to replace the flow variable with the sum of the mean and fluctuating values, namely:

$$u = \bar{u} + u' \tag{3}$$

where $\bar{u}$ represents the average value over time and $u'$ represents the pulsation value. The time-averaged continuity equation, the Reynolds equation, and scalar $\phi$ time-averaged transport equation after introducing the index symbol in the tensor are as follows:

$$\frac{\partial \rho}{\partial t} + \frac{\partial}{\partial x_i}(\rho u_i) = 0, \tag{4}$$

$$\frac{\partial}{\partial t}(\rho u_i) + \frac{\partial}{\partial x_j}(\rho u_i u_j) = -\frac{\partial p}{\partial x_i} + \frac{\partial}{\partial x_j}\left(\mu \frac{\partial u_i}{\partial x_j} - \rho \overline{u_i' u_j'}\right) + S_i, \tag{5}$$

$$\frac{\partial(\rho\phi)}{\partial t} + \frac{\partial(\rho u_j \phi)}{\partial x_j} = \frac{\partial}{\partial x_j}\left(\Gamma \frac{\partial \phi}{\partial x_j} - \rho \overline{u_j' \phi'}\right) + S. \tag{6}$$

### 2.2. Turbulence Model

The SST $k$-$\omega$ turbulence model was proposed by Menter, which considers the transport of the main shear stress in the boundary layer against the pressure gradient and can be used for the flow against the pressure gradient [13]. It is based on Bradshaw's assumption that the main shear stress is proportional to the turbulent kinetic energy, which is corrected to obtain a new turbulent viscosity. The flow equation is as follows:

$$\frac{\partial}{\partial t}(\rho k) + \frac{\partial}{\partial x_i}(\rho k u_i) = \frac{\partial}{\partial x_j}\left(\Gamma_k \frac{\partial k}{\partial x_j}\right) + G_k - Y_k + S_k, \tag{7}$$

$$\frac{\partial}{\partial t}(\rho \omega) + \frac{\partial}{\partial x_i}(\rho \omega u_i) = \frac{\partial}{\partial x_j}\left(\Gamma_\omega \frac{\partial \omega}{\partial x_j}\right) + G_\omega - Y_\omega + D_\omega + S_\omega \tag{8}$$

where $G_k$ is the turbulent kinetic energy generated by the average velocity gradient; $G_\omega$ is the $\omega$ equation; $\Gamma_k, \Gamma_\omega$ are the diffusion terms of $k$ and $\omega$; $Y_k, Y_\omega$ are the divergence terms of $k$ and $\omega$ caused by turbulence; $D_\omega$ is the orthogonal divergence term; and $S_k, S_\omega$ are user-defined source terms.

### 2.3. Vortex-Induced Vibration Related Parameters

The shape of the generation and shedding of vortices when the fluid passes through the tail of the structure is related to $Re$, and its expression is as follows:

$$Re = \frac{\rho U D}{\mu} = \frac{U D}{v} \tag{9}$$

where $Re$ is the Reynolds number, $\rho$ is the fluid density whose value is 1.185 kg/m$^3$, $U$ is the flow velocity of the fluid, $D$ is the characteristic size of the structure, that is, the outer diameter of the pipe, and $\mu$ is the dynamic viscosity coefficient.

When the fluid passes through the structure, a lateral lift force $F_L$ and a downstream drag force $F_D$, and then the dimensionless lift coefficient $C_L$ and drag force coefficient $C_D$ are obtained:

$$C_L = \frac{F_L/A}{\rho U^2/2}, C_D = \frac{F_D/A}{\rho U^2/2}. \tag{10}$$

where $A$ is the cylinder projected area perpendicular to the direction of motion. According to the $C_L$ curve, the vortex shedding frequency $f_s$ can be obtained, and the Strouhal number $St$ can be obtained by dimensionless processing of $f_s$, namely:

$$St = \frac{f_s D}{U}. \tag{11}$$

According to the DNV specification, the conditions for the occurrence of transverse vortex-induced vibration [14] are:

$$\frac{0.8}{St} < U_R < \frac{1.6}{St}, U_R = \frac{U}{f_n D} \tag{12}$$

where $U_R$ is the reduced speed, and $f_n$ is the first-order natural frequency of the pipe. The minimum wind speed at which cross-flow vortex-induced vibration occurs can be obtained as:

$$U_{min} = \frac{0.8 f_n D}{St}. \tag{13}$$

## 3. Modeling and Analysis of Bare Pipe

### 3.1. Computational Domain and Boundary Conditions

The dimensions of the unilateral restraint pipe were taken as follows: the pipe length was 17.942 m, the pipe diameter was 0.762 m, and the wall thickness was 0.019 m. The calculated first-order natural frequency was 2.355 Hz, and the estimated $U_{min}$ was 7.18 m/s. The calculation domain was set as follows: the distance of the pipe was 10 D from the entrance, 30 D from the exit, and 10 D from the upper and lower boundaries. The length of the pipe was $\pi$ D, and the polygonal unstructured mesh was divided. The polyhedral mesh technique is an advanced and mature mesh type that is used for the mesh topology of the fluid domain. Polyhedral mesh not only has the same calculation accuracy as hexahedral mesh, but also has better convergence and less mesh dependence than tetrahedral mesh [15,16]. The boundary layer was divided into 15 layers near the wall of the pipe, as shown in Figures 1 and 2.

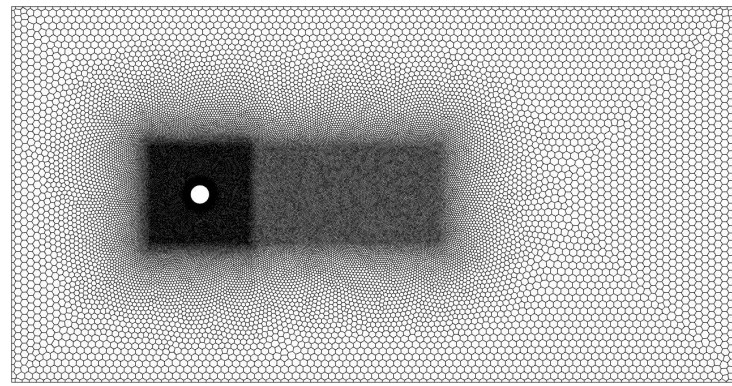

**Figure 1.** Computational domain grid.

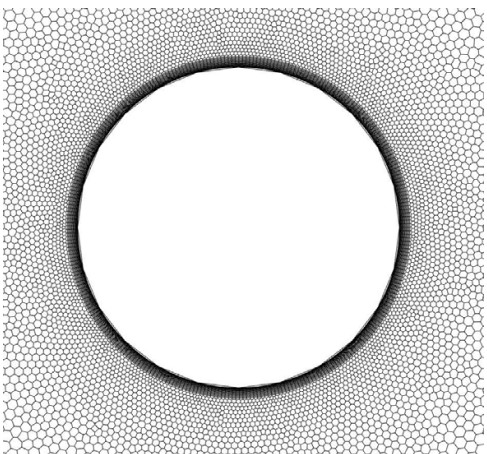

**Figure 2.** Near-wall grid of pipe.

In this study, the SST $k$-$\omega$ turbulence model was calculated with Fluent software, and the boundary conditions are shown in Table 1.

**Table 1.** Boundary condition settings.

| Location | Boundary Conditions |
|---|---|
| Left | Velocity inlet, the velocity magnitude is set to wind speed |
| Right | Pressure outlet, the gauge pressure is set to 0 |
| Up, down, front, and back | Symmetry |
| Circular cylinder | No slip wall |

The calculation method adopted the SIMPLE method [17], the pressure discrete term adopted the second-order precision, the momentum equation adopted the second-order upwind term. The transient formulation adopted the second-order implicit, initialized with the inlet boundary.

### 3.2. Calculation Result Verification

Since *St* has no definite value in the critical Reynolds number interval [18], according to the empirical formula for predicting *St* through Re in the literature [19], when $Re = 1.5 \times 10^5 \sim 3.4 \times 10^5$, *St* satisfies the following formula:

$$St = 0.1848 + 8.6 \times 10^{-4} \times \left(Re / \left(1.5 \times 10^5\right)\right)^{4.6} \tag{14}$$

when the wind speed is 6 m/s, $R_e = 3.126 \times 10^5$, which conforms to the interval required by the above formula. The theoretical value of *St* calculated by Equation (14) under this condition should be 0.21. Comparing the *St* calculated by the numerical model with the theoretical value can verify the validity of the model. According to the work of Pietro Catalano et al., a non-dimensional time-step $\Delta t U_\infty / D$ of 0.01 was used and the simulation was run for 300 time units [20]. The purpose of controlling the total number of grids is achieved by adjusting the number of circumferential grids of the pipe, thereby verifying the grid independence. The calculation results of different grid numbers are shown in Table 2.

**Table 2.** Comparison of calculation results with different grid numbers.

| Case | Number of Grids | Circumferential Nodes | $f_s$/Hz | $St$ | $\overline{C_D}$ | $C_{Lrms}$ |
|---|---|---|---|---|---|---|
| A | $440 \times 10^4$ | 240 | 1.76 | 0.224 | 0.707 | 0.287 |
| B | $510 \times 10^4$ | 300 | 1.72 | 0.218 | 0.726 | 0.333 |
| C | $600 \times 10^4$ | 360 | 1.7 | 0.216 | 0.712 | 0.330 |

In Table 2, $f_s$ is the vortex shedding frequency, $\overline{C_D}$ is the mean value of the drag coefficient, and $C_{Lrms}$ is the root mean square value of the lift coefficient. By comparison, we found that for the cases B and C, the maximum error was only 2%, and the error between the *St* of case B and the theoretical value was 3.8%; this justified the choice of model, grid, and time step. After considering calculation time and accuracy, 5.1 million grids (case B) was finally selected for subsequent analysis.

The parameters of all the cases involved in the paper are shown in Table 3.

**Table 3.** Parameter comparison of all the cases.

| Cases | Wind Speed/(m/s) | Number of Dampers | Number of Rows of Dampers |
|---|---|---|---|
| 1 | 6 | 0 | / |
| 2 | 7 | 0 | / |
| 3 | 8 | 0 | / |
| 4 | 10 | 0 | / |
| 5 | 12 | 0 | / |
| 6 | 8 | 8 | 1 |
| 7 | 8 | 8 | 2 |
| 8 | 10 | 8 | 1 |
| 9 | 12 | 8 | 1 |

The vortex shedding frequency, Strouhal number, lift coefficient, and drag coefficient of the pipe under different wind speed conditions were calculated, and the results are shown in Table 4.

**Table 4.** Comparison of calculation results of different wind speeds.

| Cases | $f_s$/Hz | St | $\overline{C_D}$ | $C_{Lrms}$ |
|---|---|---|---|---|
| 1 | 1.72 | 0.218 | 0.726 | 0.333 |
| 2 | 2.07 | 0.225 | 0.721 | 0.292 |
| 3 | 2.48 | 0.236 | 0.717 | 0.298 |
| 4 | 3.17 | 0.241 | 0.707 | 0.288 |
| 5 | 3.95 | 0.251 | 0.656 | 0.271 |

As seen in Table 4, as the wind speed increased, *St* also increased, while the drag and lift coefficient decreased accordingly. When the wind speed was 8 m/s, $f_s$ was close to the first-order natural frequency $f_n$ of the pipe, and the structure was prone to wind vortex-induced resonance. Therefore, the wind speed of 8 m/s was selected for subsequent calculation and analysis.

## 4. Modeling and Analysis of Pipe with Vibration Dampers

The vibration damper was an FD-type tuning fork structure with a total length of 0.52 m and a height of 0.07 m. Since the size of the stranded wire and angle steel was negligible compared to the size of the pipe, only the vibration damper body was retained for modeling. The damping ratio was increased by installing vibration dampers to achieve the minimum stability parameters required for the pipe. The required number of vibration dampers was calculated according to the OTC-6174. A total of eight vibration dampers were installed, and three cases were considered and compared. For case 3, the vibration dampers were not installed on the pipe. For case 6, eight vibration dampers were installed on the windward side, the leeward side, the upper side, and the lower side of the middle position of the pipe, as shown in Figure 3a. For case 7, the eight vibration dampers were installed in two rows, which were symmetrical about the midpoint of the pipe, and the distance between the vibration dampers of different rows was 0.1 m, as shown in Figure 3b. The number of grids in case 6 and case 7 was 7.38 million, and other settings were the same as those without the vibration dampers. The following calculation and analysis were performed for the wind speed of 8 m/s.

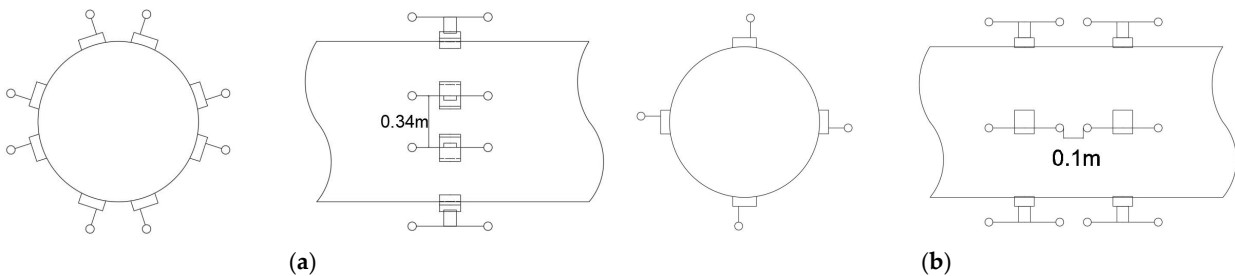

(**a**)                                              (**b**)

**Figure 3.** Arrangement of the vibration dampers. (**a**) The eight vibration dampers are installed in one row. (**b**) The eight vibration dampers are installed in two rows.

### 4.1. Lift and Drag Force Analysis

Figure 4 shows the lift and drag force coefficient curves and the corresponding amplitude spectra of the three cases.

(**a**)

(**b**)

(**c**)

(**d**)

**Figure 4.** The lift, drag coefficient curves and amplitude spectra. (**a**) Lift coefficient curves of cases 3, 6, and 7; (**b**) amplitude spectra of lift coefficients; (**c**) drag coefficient curves of cases 3, 6, and 7; (**d**) amplitude spectra of drag coefficients.

Figure 4a gives the lift coefficient curves of three cases. The results indicated that the maximum value after stabilization was about 0.4 without the vibration dampers, and the maximum value was reduced to about 0.15 after the vibration dampers were installed. The root mean square of lift coefficient values corresponding to case 3, 6, and 7 were 0.298,

0.103, and 0.105, respectively, and the root mean square of lift coefficient was reduced by about 65% after installing the vibration dampers. The lift coefficient curves of the three cases fluctuated up and down around the zero value, showing periodic changing. After the vibration dampers were installed, the lift coefficient curves entered a stable state faster. The frequency of the lift coefficient of case 6 was increased by 3.23% compared with that of case 3, and the frequency of the lift coefficient of case 7 was reduced by 16.13% compared with that of case 6, as shown in Figure 4b. In addition, the Strouhal numbers corresponding to the three cases were 0.236, 0.244, and 0.198, respectively.

Figure 4c gives the drag coefficient curves of the three cases. The results demonstrated that the drag coefficient of case 3 was the largest, with a mean of 0.717 and a SD (standard deviation) of 0.014. The second largest was case 7, with a mean of 0.712 and a SD of 0.002. The drag coefficient of case 6 was the smallest, with a mean of 0.698 and a SD of 0.003, a decrease of 2.65% relative to case 3. The fluctuation range of the drag coefficient was weakened after the vibration dampers were installed, and the reduction of the drag coefficient in case 6 was more obvious than that in case 7. As shown in Figure 4d, the frequency of the drag coefficient in the three cases was around twice the frequency of the lift coefficient of the corresponding case, and the frequency relationship between the drag force and the lift force did not change.

### 4.2. Analysis of Different Wind Speed

According to the comparison in Section 4.1, both installation methods effectively reduced the lift coefficient of the pipe, but case 6 had a certain inhibitory effect on the drag coefficient. On this basis, the effects of installing the vibration dampers on various parameters under three wind speeds were calculated. The calculation results are shown in Table 5.

**Table 5.** Comparison of results after installing vibration dampers at different wind speeds.

| Cases | $f_s$/Hz | $St$ | $\overline{C_D}$ | $C_{Lrms}$ |
|-------|---------|-------|---------|-----------|
| 6 | 2.56 | 0.244 | 0.698 | 0.103 |
| 8 | 3.20 | 0.244 | 0.683 | 0.096 |
| 9 | 3.87 | 0.246 | 0.622 | 0.089 |

Compared with the results of the same wind speed in Table 4, the influence of the installation of the vibration dampers on the vortex shedding frequency increased or decreased, increasing by 3.23% when the wind speed was 8 m/s, and decreasing by 2.03% when the wind speed was 12 m/s. The impact of the vibration dampers on the drag coefficient was also obvious; the drag coefficient was reduced by 3.65%, 3.39%, and 5.18%, respectively, in cases 3, 6, and 7. The vibration dampers had the greatest influence on the lift coefficient. The lift coefficients decreased by 65.44%, 66.67%, and 67.16%, respectively, in the three cases. The influence of the vibration dampers on the lift coefficient and drag coefficient tended to increase with an increase in wind speed.

### 4.3. Flow Field Analysis

Figures 5–7 show the contours of cases 3, 6, and 7, including the velocity contours and the pressure contours, and we then analyzed the influence of the vibration dampers on the flow field around the pipe.

According to the velocity contours, the obvious Karman vortex street phenomenon was observed. In the velocity contour of the three cases, a circular low-velocity area appeared on the windward side of the pipe. Compared with case 3, after the vibration dampers were installed the backflow phenomenon occurred behind the vibration dampers. The high-velocity areas on the upper and lower sides of the pipe were destroyed by the vibration dampers, and the backflow range in the wake area increased, causing the distribution of the high-velocity area to also shift backward, as shown in Figures 6a and 7a.

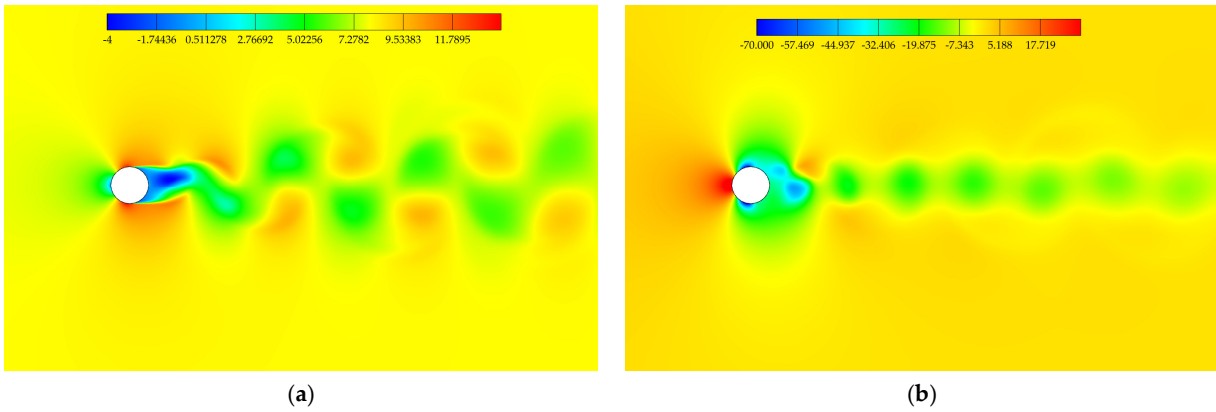

**Figure 5.** Contour diagrams of case 3: (**a**) velocity contour; (**b**) pressure contour.

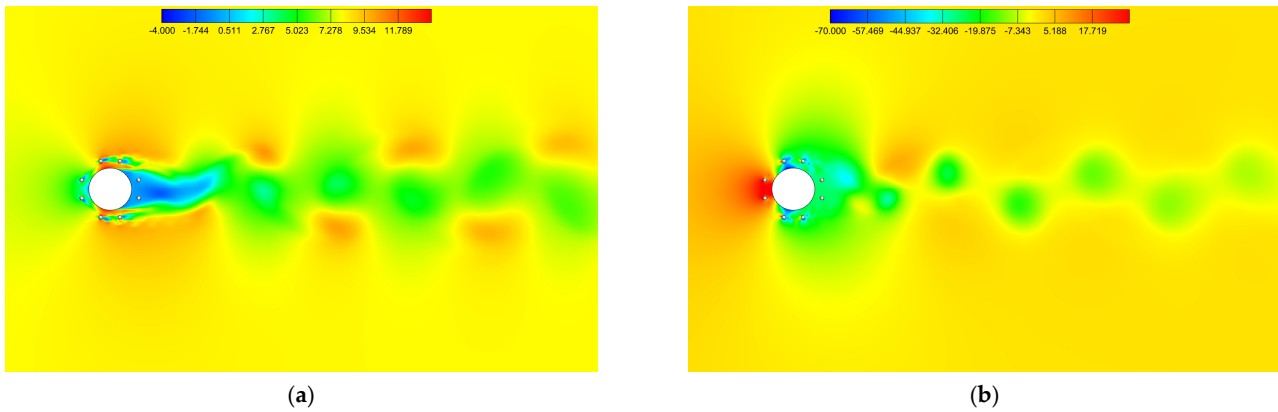

**Figure 6.** Contour diagrams of case 6: (**a**) velocity contour; (**b**) pressure contour.

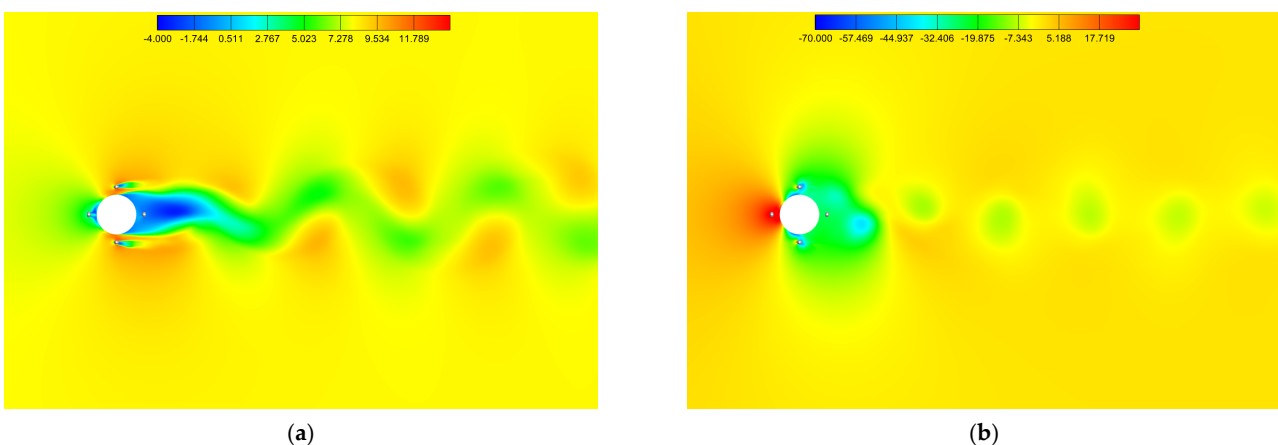

**Figure 7.** Contour diagrams of case 7: (**a**) velocity contour; (**b**) pressure contour.

According to the pressure contours, when the wind flowed cross the pipe, there was a positive high-pressure area on the windward side of the pipe, and there were negative pressure areas on the upper and lower sides and the leeward side, and the upper and lower sides were negative high-pressure areas. In the wake area, positive and negative pressure areas appeared alternately, and corresponded to each other but in opposite directions to the velocity contour. The pressure distribution around the vibration dampers was similar to that of the round pipe. The pressure on the leeward side was smaller than that on the windward side, and the negative high-pressure area of the entire structure increased. The pressure in the wake area of the pipe with the vibration dampers was generally lower than

the pressure of the pipe without the vibration dampers. In addition, the farther from the circular pipe in the wake area, the closer the pressure was to the overall flow field, as shown in Figures 6b and 7b.

Figures 8–10 show the three-dimensional vorticity of cases 3, 6, and 7, plotted using the Q criterion [21]. When the wind flowed cross the pipe, the wake region shed vortices in opposite directions and alternated periodically, as shown in Figure 8. In case 6, the vibration dampers disturbed the vortex in the wake area of the pipe, and the first pair of alternating up and down vortices in the wake area were severely damaged, resulting in many small vortices around the pipe. Compared with case 3, the vortex shedding moved backward as a whole, as shown in Figure 9, but owing to the smaller size of the vibration dampers compared to the pipe and the concentrated installation, there was no obvious effect on the vortex shedding behind. In case 7, the small vortices around the pipe were not as dense as those in case 6, but the vortices shedding alternately in the wake region along the length of the pipe changed more obviously, and the vortex shedding directly behind the vibration dampers occurred obviously later than that without the vibration dampers, thereby reducing the lift coefficient.

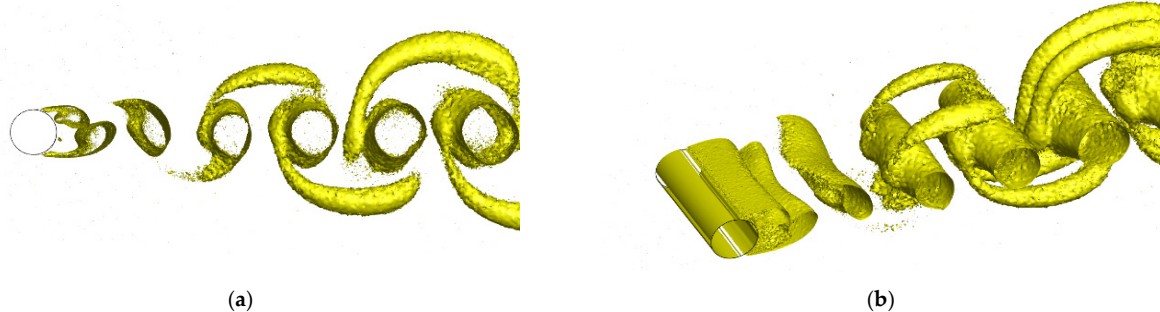

(**a**)                                    (**b**)

**Figure 8.** Three-dimensional vorticity diagram of case 3: (**a**) main view; (**b**) global view.

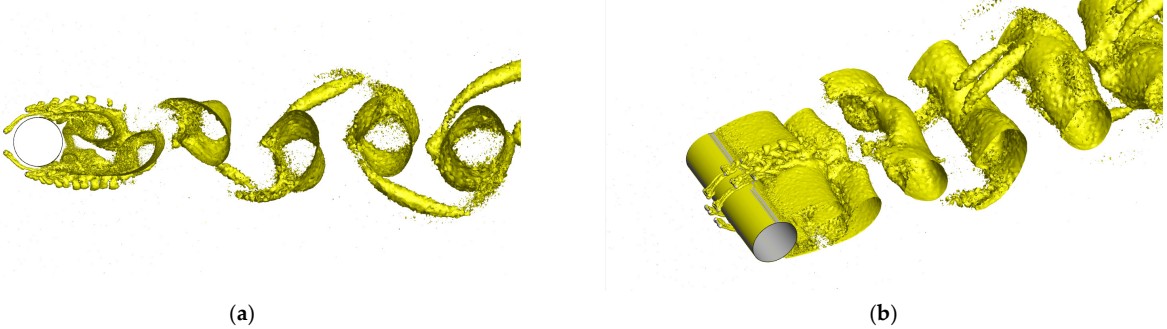

(**a**)                                    (**b**)

**Figure 9.** Three-dimensional vorticity diagram of case 6: (**a**) main view; (**b**) global view.

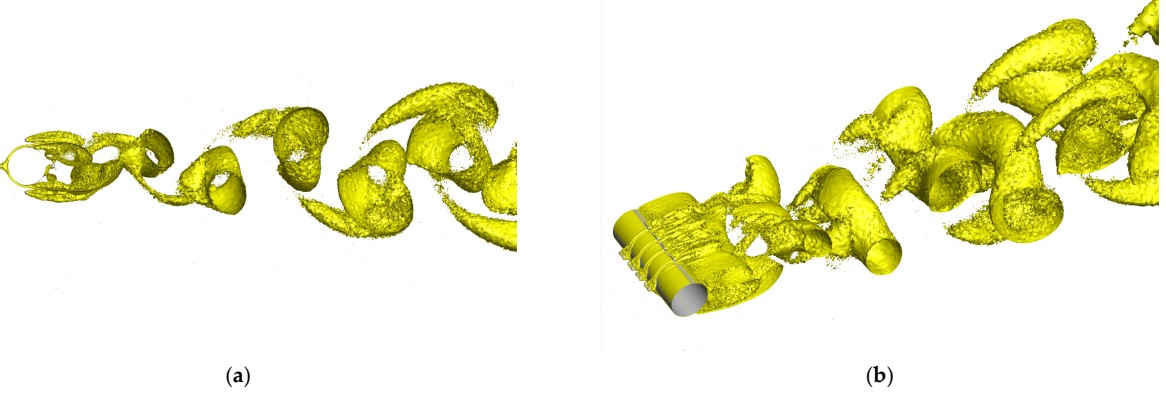

(**a**)                                    (**b**)

**Figure 10.** Three-dimensional vorticity diagram of case 7: (**a**) main view; (**b**) global view.

## 5. Conclusions

In this paper, the vortex-induced force of the deep-water jacket pipe was studied based on the CFD method. The vibration dampers commonly used for vibration reduction of power lines were used on the jacket pipes. The models of the pipe with the vibration dampers were established, and the stability of the model was verified. The wind speed range causing the wind-induced VIV of the pipe was preliminarily estimated. The influence of installing the vibration dampers on the lift and drag coefficients of the pipe was analyzed, and we explained the reason from the perspective of the flow field. The main conclusions are as follows:

(1) The installation of vibration dampers can effectively reduce the vortex-induced force. The lift coefficient of the pipe is reduced by about 65%, the drag coefficient of the pipe is slightly reduced.

(2) The influence of the vibration dampers on the vortex-induced force is stable under different wind speeds. The error of the influence of the vibration dampers on the vortex-induced force under different wind speeds is about 2%.

(3) The influence of the vibration dampers on the vortex-induced force is related to the arrangement. Compared with the arrangement in one row, when the vibration dampers are arranged in two rows, the vortex shedding frequency is reduced by 16%.

(4) The influence of installing the vibration dampers on the flow field of the pipe was analyzed. The high-velocity areas on the upper and lower sides of the pipe are destroyed after the vibration dampers are installed. The vortex shedding is destroyed, and the overall vortex shedding moves backward, thereby reducing the wind-induced vortex force.

The jacket will have many pipes affected by vortex-induced vibration. This paper only considered one pipe that will vibrate during the construction stage and did not consider different pipe diameters. In addition, fluid–structure interaction was not considered, which requires future study.

**Author Contributions:** Conceptualization, C.L. and L.L.; methodology, C.L. and L.L.; software, Z.W., J.C. and Y.Y.; validation, L.L., C.L. and Z.W.; formal analysis, C.L.; investigation, Z.W.; resources, C.L.; data curation, J.C. and Y.Y.; writing—original draft preparation, C.L., Z.W. and J.C.; writing—review and editing, L.L. and Y.Y.; supervision, C.L. and J.C.; project administration, C.L. and L.L. All authors have read and agreed to the published version of the manuscript.

**Funding:** This research received no external funding.

**Institutional Review Board Statement:** Not applicable.

**Informed Consent Statement:** Not applicable.

**Data Availability Statement:** Not applicable.

**Conflicts of Interest:** The authors declare no conflict of interest.

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
