# Peer review of "Influence of Vibration Dampers on the Vortex-Induced Force and Flow Characteristic of Deep-Water Jacket Pipe"

_applsci, doi:10.3390/app122010219_

Round 1

Reviewer 1 Report

The authors present a method to reduce VIV during the assembly of Deep water Jacket pipes using 8 vibration dampers fitted on one ore two sections, They used CFD with 5.1 mio. elements. They should declare whether they use a commercial or a home-made code. They conclude that that the dual-section arrangement of dampers is effective reducing VIV.

Reviewer 2 Report

Please consider the following comments in the revised manuscript:

-         SST kω turbulence model was used. Why? Why not LES? Can SST kω turbulence model simulate such problem accurately? Please discuss.

-         Please revise Eq. 10 by considering the following paper:

Comment on ‘‘Summary of frictional drag coefficient relationships for spheres: Evolving solution strategies applied to an old problem”

-         Please present some evidence for showing independency of results to computational domain grid

-         The discussion on the results were not presented appropriately. Why such results were achieved? Please discuss.

Reviewer 3 Report

The manuscript concentrates on the suppression of VIV by using dampers. I recommend to  highlight this key topic in the title and abstract. Normally the phenomenon is referred to as vortex-induced-vibration (VIV), the phrase ‘wind-vortex-induced’ used by the manuscript is really wired. The most confusing point is that the authors were describing the vibration of the pipe (e.g. the governing equation and the resultant displacement, vibrating frequency of the structure), even though the study is all about VIV. Indeed, VIV scenario and the suppression strategies have been extensively studied over the past decades. The novelty of current study is not clear.  The manuscript is poorly written, the language needs to be polished carefully.  The results are badly presented. I do not support the acceptance of current version.

Round 2

Reviewer 2 Report

The authors did not consider my previous comments (1,2 and especially 4), appropriately. However, the manuscript can be accepted.